

# Developing drought impact functions for drought risk management

Sophie Bachmair[1], Cecilia Svensson[2], Ilaria Prosdocimi[2,3], Jamie Hannaford[2], and Kerstin Stahl[1]

[1]Environmental Hydrological Systems, Faculty of Environment and Natural Resources, University of Freiburg, Freiburg, 79098, Germany
[2]Centre for Ecology and Hydrology, Wallingford, UK
[3]Now at the Department of Mathematical Sciences, University of Bath, Claverton Down, Bath, Somerset, BA2 7AY, UK

*Correspondence to*: Sophie Bachmair (sophie@bachmair.info)

**Abstract.** Drought management frameworks are dependent on methods for monitoring and prediction, but quantifying the hazard alone is arguably not sufficient; the negative consequences that may arise from a lack of precipitation must also be predicted if droughts are to be better managed. However, the link between drought intensity, expressed by some hydro-meteorological indicator, and the occurrence of drought impacts has only recently begun to be addressed. One challenge is

the paucity of information on ecological and socio-economic consequences of drought. This study tests the potential for developing empirical "drought impact functions" based on drought indicators (Standardized Precipitation and Standardized Precipitation Evaporation Index) as predictors, and text-based reports on drought impacts as a surrogate variable for drought damage. While there have been studies exploiting textual evidence of drought impacts, a systematic assessment of the effect of impact quantification method and different functional relationships for modeling drought impacts is missing. Using South-

East England as a case study we tested the potential of three different data-driven models for predicting drought impacts quantified from text-based reports; logistic regression, zero-altered negative binomial regression ("hurdle model"), and an ensemble regression tree approach ("random forest"). The logistic regression model can only be applied to a binary impact/no impact time series, whereas the other two models can additionally predict the full counts of impact occurrence at each time point. While modeling binary data results in the lowest prediction uncertainty, modeling the full counts has the

advantage of also providing a measure of impact severity, and the counts were found to be predictable within reasonable limits. However, there were noticeable differences in skill between modeling methodologies. For binary data the logistic regression and the random forest model performed similarly well based on leave-one-out cross-validation. For count data the random forest outperformed the hurdle model. The between-model differences occurred for total drought impacts as well as for two subsets of impact categories (water supply and freshwater ecosystem impacts). In addition, different ways of

defining the impact counts were investigated, and were found to have little influence on the prediction skill. For all models we found a positive effect of including impact information of the preceding month as a predictor in addition to the hydro-meteorological indicators. We conclude that, although having some limitations, text-based reports on drought impacts can





provide useful information for drought risk management, and our study showcases different methodological approaches to developing drought impact functions based on text-based data.

## 1 Introduction

Drought is a major natural hazard with manifold impacts on the environment, the economy and wider society. While the hazard itself can rarely be avoided, drought risk assessment and management are important tools for responding to the hazard to mitigate impacts and for proactively planning for future droughts (Wilhite et al., 2000). Risk is commonly understood as a combination of the probability of an event and its negative consequences (UNISDR, 2009). Hence, it is not only important to better understand and predict the hazard, but also the likely consequences of the hazard, which depend on the vulnerability of the exposed people and assets at risk. Much research on drought has focused on characterizing the hazard (Briffa et al., 1994; McKee et al., 1993; Stagge et al., 2015a), and less on drought impacts (Bachmair et al., 2016a; Naumann et al., 2015). Also, most drought early warning systems monitor and/or forecast the hazard but do not provide information on when and where a precipitation deficit may turn into negative consequences. In a review of flood risk assessment, the authors state that hazard assessment receives much more attention than the assessment of negative consequences or damage, which "is treated as some kind of appendix within the risk analysis" (Merz et al., 2010). In comparison to drought, however, there have been considerable efforts to assess and model flood damage (e.g. Jongman et al., 2012; Merz et al., 2013; Schröter et al., 2014; Spekkers et al., 2014; Thieken et al., 2005).

A common approach for assessing the negative consequences of natural hazards is the use of damage functions, variously called vulnerability functions or stage-damage-curves depending on the damage variable used and on author conventions (e.g. Michel-Kerjan et al., 2013; Papathoma-Köhle et al., 2015; Tarbotton et al., 2015). Such damage functions are usually continuous curves relating the hazard intensity (e.g. inundation depth or wind velocity) to the negative effects of the hazard, often expressed as a damage ratio of buildings. Transferring the concept of (empirical) damage functions to drought risk assessment presents many challenges, and has only recently begun to be addressed (Naumann et al., 2015). The main challenges can be conceptualized as follows: first, what is a suitable indicator characterizing the drought hazard (abscissa in Figure 1a)? Drought is known as a multi-dimensional hazard affecting different domains of the hydrological cycle and with different response times (Wilhite and Glantz, 1985). Second, what is a suitable damage variable for drought effect/damage (ordinate in Figure 1a)? This is particularly challenging given that many negative consequences of drought, hereafter drought impacts, are non-structural and hard to quantify or monetize (e.g. local water supply shortages or restrictions on domestic water use, impaired navigability of streams, or ecological impacts such as irreversible deterioration of wetlands or fish kills). Also, there is a paucity of drought impact data with sufficient spatial and temporal resolution except for the agricultural sector (Bachmair et al., 2016b). The third challenge is identifying an adequate functional relationship for relating hazard intensity to a damage variable (red lines in Figure 1a).





Regarding the first challenge (hazard intensity variable), several authors have empirically assessed which drought indicators are best linked to certain drought impact types such as, for example, vegetation stress (e.g. Bachmair et al., 2016a; Blauhut et al., 2016; Lorenzo-Lacruz et al., 2013; Stagge et al., 2015b; Stahl et al., 2012; Vicente-Serrano et al., 2013). These drought indicators tend to be measures of hydro-meteorological variables which are relatively easy to quantify objectively, such as,

for example, rainfall. Regarding the second challenge (drought damage variable), studies include a variety of data types representing the drought impact, including crop yield (e.g. Hlavinka et al., 2009; Naumann et al., 2015; Potopová et al., 2015; Quiring and Papakryiakou, 2003); wildfire occurrence (e.g. Gudmundsson et al., 2014); drought-induced building damage (Corti and Wüest, 2011); and hydropower production (Naumann et al., 2015). While the above data relate to one specific type of drought impact, text-based reports on drought impacts as assembled by the US Drought Impact Reporter

(DIR) (Wilhite et al., 2007) and the European Drought Impact report Inventory (EDII) (Stahl et al., 2016) provide information on different types, including indirect and non-market impacts (e.g. ecological impacts, impacts on human health). However, for empirical damage functions such qualitative data needs to be quantified, although this transformation inevitably introduces uncertainties. A few studies exploited text-based impact reports from the EDII by converting them into binary time series of impact occurrence (Blauhut et al., 2015b, 2016; Stagge et al., 2015b). Building on these efforts,

Bachmair et al. (2015, 2016a) derived the number of impacts based on text-based data, providing a surrogate measure of impact severity. The suitability of these different impact quantification methods has not yet been systematically assessed.

Regarding the third challenge, different data-driven models have been deployed depending on the probability distribution of the drought impact variable, and the relation with the hazard indicator (e.g. linear regression, logistic regression, power law functions (e.g. Blauhut et al., 2015b; Naumann et al., 2015)). In addition to parametric models, non-parametric approaches

such as classification and regression trees have been successfully applied for flood damage modeling (Merz et al., 2013; Spekkers et al., 2014). While an ensemble regression tree approach has also been tested for modeling text-based drought impacts (Bachmair et al., 2016a), assessing the performance of different  functional relationships remains an unmet challenge.

The aim of this study is to develop empirical "drought impact functions" based on text-based reports from the EDII as

surrogate information on drought damage, and thereby assess possibilities and limitations of transferring the concept of damage functions to drought. Specifically, we test

- the effect of different methods of quantifying text-based drought impact information, and
- the predictive power of three data-driven models for linking drought intensity with drought impacts.

We use a selection of standardized hydro-meteorological indices as drought hazard indicators.





## 2 Data

### 2.1 Study area

We selected South-East England (SSE) as a case study for developing the drought impact functions (Figure 1b). This is a level 1 region of the Nomenclature of Units for Territorial Statistics (NUTS1), a spatial unit used in the European Union.

The reasons for choosing SEE include the good data availability in the EDII for this region, and the importance of drought risk assessment for this area given the severe droughts that have occurred in south-eastern UK in the past (e.g. Kendon et al., 2013; Marsh, 2007). The south-east is one of the driest parts of the UK, but with some of the highest water demands. The region hosts a large population, approximately nine million (Office for National Statistics, 2016), and high concentrations of commercial and industrial activity. Consequently, parts of the region are already water stressed, with pressures on the water

environment expected to increase in future (Environment Agency, 2009) . The EDII drought impacts for the SEE study area mainly consists of impacts on the water supply and on freshwater ecosystems.

### 2.2 Predictors

As candidate predictors we selected the commonly used drought indicators Standardized Precipitation Index (SPI) (McKee et al., 1993) and Standardized Precipitation Evaporation Index (SPEI) (Vicente-Serrano et al., 2010) of 1-6, 9, 12, and 24

months accumulation period (hereafter SPI-$n$ or SPEI-$n$). The SPI (SPEI) compares the total precipitation (climatic water balance) of a certain location over a period of $n$ months with its multiyear average (Vicente-Serrano et al., 2010; Zargar et al., 2011). SPI and SPEI are based on E-OBS gridded rainfall and temperature data (v12.0, 0.25° spatial resolution) (Haylock et al., 2008). We used the R package "SCI" (Gudmundsson and Stagge, 2014) for SPI and SPEI calculation (gamma distribution for SPI; generalized logistic distribution for SPEI; standardization period for both variables: 1970-2012).

Evapotranspiration was determined using the Hargreaves-Samani method (Hargreaves and Samani, 1982). As additional predictors, used to account for temporal trend and seasonality, we chose the year (Y) and the month (M, expressed as a sinusoid) of impact occurrence (Bachmair et al., 2016a). For parts of the analysis the impact data of the preceding month (section 2.2) was introduced as a further predictor to address autocorrelation of residuals. All predictor time series have monthly resolution. That is, although most of the SPI and SPEI accumulation periods are longer than a month, each index is

calculated for a moving window that is shifted one month at a time.

### 2.3 Drought impacts

Drought impact information for our SSE case study region comes from the European Drought Impact report Inventory (EDII) (Stahl et al., 2016), accessible at http://www.geo.uio.no/edc/droughtdb/ (data extraction: July 2016). The EDII contains text-based reports on drought impacts. Each report states: i) the location of occurrence (making reference to

administrative regions at different NUTS levels); ii) the time of occurrence (at least the start and end year); and iii) the type of impact (assignment to predefined impact categories and subtypes). For quantitative analysis these reports need to be



converted into time series of impact information. We tested three different approaches of impact counting to address the uncertainty associated with impact report quantification. The general procedure follows previous studies (Bachmair et al., 2015, 2016a). For our analysis monthly time series are used. Not all impact reports state the start and end month of impact occurrence; if only information about the season was available, we assumed drought impact occurrence during each month

of this season (winter= DJF, spring= MAM, summer= JJA, fall= SON). Impact reports only stating the year of occurrence, or with incomplete information about impact category or subtype, were omitted.

Impact counting methods:

1.  Only presence versus absence of drought impacts per month is considered (Blauhut et al., 2015b, 2016; Stagge et al., 2015b), resulting in binary time series of impact occurrence (hereafter $I$).

2.  All impact reports are counted. If an impact report states $n$ impact subtypes, there are $n$ impact counts for each specified month (Bachmair et al., 2015, 2016a). This results in time series of number of impact occurrences (hereafter $N_I$). For instance, for impact category "public water supply" seven impact subtypes may be specified, ranging from local water supply shortage (e.g. drying up of springs/wells, reservoirs, streams) over bans on domestic and public water use (e.g. car washing, watering the lawn/garden, irrigation of sport fields, filling of swimming pools) to increased costs/economic

losses (Stahl et al., 2016)). In total, there are 15 different impact categories in the EDII, each with its own set of subtypes.

3.  An impact report assigned to one impact category only counts once, independent of how many impact subtypes are specified. The resulting time series shows the same dynamic as for Method 2 but has lower $N_I$.

$N_I$ provides a measure of impact severity, but the information is likely more uncertain than binary data. For our analysis we

considered total impacts in SEE (all impact categories), and two different subsets: water supply impacts and impacts related to freshwater ecosystems. These two impact categories make up the dominant part of the total impacts in SEE. As a consequence of the specific counting decision as well as the dynamic nature of the EDII, to which new entries may have been added and amendments or correction to existing entries may have been made in the meantime, the time series used in this study may differ slightly from those used in previously published studies.

**3 Methods**

**3.1 Data-driven models**

To establish a functional relationship between drought indicators (and further predictors) and drought impacts $I$ or $N_I$, we tested three different models:

1)  logistic regression (LG) for the presence or absence of impact data as a binary response variable (Blauhut et al.,
2015b, 2016; Stagge et al., 2015b);





2) zero-altered negative binomial regression; this parametric model for count data is also known as a "hurdle" model (HM) (Zeileis et al., 2008); and

3) a "random forest" (RF) model (Breiman, 2001), which is an ensemble of regression trees.

Logistic regression was selected because it has been previously used for drought impact modeling (Blauhut et al., 2015b;

Gudmundsson et al., 2014). For modeling count data we aimed to explore the predictive power of one parametric model and a non-parametric alternative. Since the impact data contains many zeros, we selected the hurdle model, which is capable of dealing with excess zeros (Zeileis et al., 2008). The HM has been successfully applied to ecological datasets with zero-inflation (Ver Hoef and Jansen, 2007; e.g. Potts and Elith, 2006). The RF model represents a flexible machine learning approach that can handle non-linearities and predictor interactions (Breiman, 2001; Liaw and Wiener, 2002). The RF model

has been extensively used for many applications in environmental science (e.g. Bachmair et al., 2016a; Catani et al., 2013; Oliveira et al., 2012; Park et al., 2016; Valero et al., 2016).

LG belongs to the class of generalized linear models (Zuur et al., 2009a). The (logit-transformed) probability of impact occurrence ($\pi$) is modeled as a linear function of the predictors $x_i$ following Eq. (1):

$$\log\left(\frac{\pi}{1-\pi}\right) = \alpha + \sum_i \beta_i x_i$$

The left-hand side represents the logit transformation; the model parameters $\alpha$ and $\beta$ are estimated by maximum likelihood (McCullagh and Nelder, 1989).

The HM consists of two parts: a hurdle part for modeling zero versus larger counts, and a truncated count part for modeling positive counts (Zeileis et al., 2008). We selected a binomial model with logit link for the hurdle part (see LG); since the impact data is over-dispersed (variance larger than theoretically expected, in this case larger than the mean) we selected a

negative binomial model for the count part with log link. For details of this model see Zeileis et al. (2008) and Zuur et al. (2009b). We used the R package "pscl" for the implementation (Jackman, 2015).

The RF model is a machine learning algorithm where a large number of regression trees are grown on bootstrapped subsamples of the data (Breiman, 2001). We used the R package "randomForest" (Liaw and Wiener, 2002). The default values were kept for all model parameters; the variable $n_{tree}$ was set to 1000. Details about drought impact modeling using

RF can be found in Bachmair et al. (2016a). For this study, however, we found that results are best when applying a square root transformation to the response variable for the binary part of the time series, and no transformation for the count part. We obtain the final modeled time series by running the RF model twice with: a) square root transformed data, and b) untransformed data. The back-transformed output from model a) is replaced with the output from model b) if the modeled number of impacts from a) is >=1. Raw residuals refer to the difference between this final modeled time series and observed

data.





### 3.2 Modeling approach

The predictors for LG and HM were selected using stepwise regression (backward and forward selection with the Bayesian Information Criterion as the selection criterion (Schwarz, 1978)). The models contain an intercept and linear terms for each predictor. For the two-part HM we only kept significant predictors (p < 0.05) for each model part in case a predictor was

identified as important for one part, yet not for the other. A further criterion was applied when the cross-correlation between two predictors exceeded 0.7. To avoid co-linearity between the predictors, only the predictor showing the best correlation with the predictand (i.e. the impacts) was kept.

For RF there is no prior predictor selection; best performing predictors are identified within the algorithm. Confidence intervals for LG and HM are computed using bootstrapping (resampling with replacement). For RF, confidence intervals are

based on the predictions of all individual random forest trees; each tree is constructed based on a bootstrapped subsample containing two thirds of the data (Liaw and Wiener, 2002) .

For the analysis we used a censored time series based on years with drought impact occurrence rather than the entire time series (Bachmair et al., 2015, see 2016a). The rationale is that there may be a lack of impact reporting for certain drought events; hence we only focus on parts of the time series with reported drought impacts. All months of all years with drought

impact occurrence were selected plus an additional six months buffer before and after the drought year to include sufficient variability for model training. This resulted in n=234 months for total impacts, n=198 for water supply impacts, and n=174 for freshwater ecosystem impacts.

To assess the model's predictive power we performed leave-one-out cross-validation, i.e. each month is left out once for model training, and a prediction is made for this omitted month. We evaluated the model performance regarding its

capability of predicting binary data and count data (HM and RF). For the binary performance evaluation we rounded the time series of LG; for HM and RF, data points <1 were rounded, and data points >1 truncated to 1. We used the following performance metrics: hit rate (i.e. the proportion of predictions for which the presence or absence of impacts is correctly identified), false positive and false negative rate. The model performance metric for the count part of HM and RF is the Kling-Gupta-Efficiency (KGE), which is based on the difference in mean, standard deviation, and correlation between the

observed and the leave-one-out predicted series (Gupta et al., 2009). KGE lies between 1 (perfect fit) and negative infinity (worst fit).

### 4 Results

#### 4.1 Selected predictors

The stepwise approach (see 3.2) resulted in the following predictors being selected for the LG model: SPI-6, SPEI-24, and M

for modeling total impacts; SPI-6 and SPI-24 for water supply impacts; SPI-3, SPI-6, SPI-24, and Y for freshwater ecosystem impacts. The selected predictors for the HM are SPI-6 and SPEI-24 for the hurdle part, and SPI-6 and Y for the



count part (total impacts for both methods of impact quantification). For water supply and freshwater ecosystem impacts different predictors were automatically selected for both model parts and methods of impact quantification (water supply impacts: SPI of short, medium, and long accumulation periods; freshwater ecosystem impacts: SPI and SPEI of short, medium, and long accumulation periods, and Y). For RF, all predictor are used, yet similar predictors as for LG and HM

were identified as most important during regression tree construction.

### 4.2 Selected predictors

Figure 2 shows the dependence of the observed or modeled response variable (total impacts, $N_I$ quantified after method 3) on the selected predictors; note that only the dependence on SPI-6 and SPEI-24 is displayed although the models include further predictors (e.g. M and Y). The top panels reveal a complex relationship between drought indicators and observed $I$ or $N_I$.

Positive impact counts occur not only for negative drought indicator values: there are four instances of $I$ for positive values of both drought indicators (front left quadrant), and several data points with positive $N_I$ yet negative indicator values for only one of SPI-6 or SPEI-24. The panels showing fitted data and an additional interpolated surface to aid visualization can be regarded as a three-dimensional version of the common two-dimensional damage functions based on one predictor. For LG, the fitted data reveal a comparably smooth increase of the likelihood of impact occurrence from positive to negative values

for both selected drought indicators. For HM and RF, the response surface is more rugged. The RF model better captures observed $N_I$ than HM, especially for cases with negative SPEI-24 but less negative SPI-6; HM strongly underestimates these $N_I$. Figure 3 additionally shows time series of observed versus fitted $I$ or $N_I$ and confidence intervals. Both count data models tend to underestimate medium to high $N_I$. HM additionally shows estimates of impact occurrence when none occurred. The confidence intervals for LG and HM are rather narrow, whereas they are wider for RF. Note that for the impact

quantification method 2 (same dynamics but higher $N_I$), the underestimation of high $N_I$ by RF is less pronounced, whereas it is much more pronounced by HM (not shown).

An analysis of the residuals revealed significant autocorrelation up to a lag of 8 months depending on the model and impact quantification method (see examples in Figure 4). For RF, the autocorrelation of the residuals is less pronounced than for LG and HM. To take the autocorrelation into account, impact information for the preceding month was included in the model.

For the binary part of the model, this meant whether or not impacts occurred in the preceding month. For the counts part, the number of impacts in the preceding month was added as a predictor. The inclusion of this autoregressive part in the model generally resulted in a considerable decrease in the autocorrelation of the residuals. For HM, however, it also caused significant overprediction of $N_I$ for two data points.

### 4.3 Selected predictors

For each of the different models, the predicted series from the leave-one-out cross-validation was compared with the observed series. The evaluation of the predictive performance considering binary data and count data (HM and RF) separately yielded the following findings:





1) noticeable differences between models,

2) small differences between impact counting methods (i.e. all types of response data are equally well predicted),

3) a positive effect of including impact information of the preceding month as an additional predictor, and

4) similar results regarding between-model differences for different impact subtypes.

Generally, for binary data, LG and RF perform similarly well with a hit rate of roughly 0.8; the hit rate of the hurdle model is distinctly lower (Figure 5 columns 1-2). For count data, RF is superior to HM. The temporal dynamics of $N_I$ are better reproduced by RF than HM (see Figure 6). However, underprediction of higher impact counts for the RF model lead to a lower mean and standard deviation than observed, resulting in KGE values less than 0.6. The HM shows an even stronger underprediction of high $N_I$ and frequent impact occurrence predictions despite absent impacts, resulting in KGE values less

than 0.4. The impact quantification method (Figure 5 column 1 vs. 2) has hardly any effect on RF performance for either binary or count data. For HM, counting method 3 (lower $N_I$) leads to a small but notable increase in performance.

For all the models there is a generally positive effect of including impact information from the preceding month (Figure 5 column 1 vs. 3). The hit rate of LG and RF increases to > 0.9, and KGE values increase by ca. 20 percent. For HM, however, strong overestimation can be noticed for summer 2006 (Figure 5). When subsetting the total impacts on water supply and

freshwater ecosystems, respectively, the same general picture of between-model differences as for total impacts is seen. That is, RF and LG are similar regarding binary data, and RF is superior to HM for the counts part (Figure 5 column 4-5). However, apart from this the results are varied. There is either a slightly increased or decreased predictive performance depending on the model, impact counting method, and binary versus count data performance metric (only impact counting method 2 is shown). Notable is a decreased performance of HM for water supply impacts, yet an increase for freshwater

ecosystem impacts, compared with the prediction of total impacts.

## 5 Discussion

Previous studies exploiting impact data from the EDII have primarily used impact occurrence information coded as a binary variable (presence versus absence of impacts). This method of impact quantification has several advantages: it is simple to implement and communicate, and contains fewer subjective decisions and lower uncertainty. However, it does not provide

information about the severity (in some quantitative sense) of the drought impacts. For characterizing drought onset and termination binary data may be sufficient. Once in drought, however, there is less possibility of identifying specific times or regions more severely affected than others. Although the number of drought impacts is undeniably more uncertain than a simple measure of presence/absence of impacts, it provides a measure of impact severity and was predictable within reasonable limits. We therefore conclude that there is value in using the number of impacts as a variable to express drought

damage. The fact that the differences between both methods of impact counting were mostly small demonstrates that either approach is useful and relatively robust. For the hurdle model, however, the method resulting in lower impact counts (only





differentiating between impact categories but not subtypes) yielded better results. Overall, we recommend interpreting impact counts as a severity metric rather than as representing the true number of observed impacts.

Testing three data-driven models revealed the superiority of RF with respect to predictive model performance. The discriminatory power of LG and the RF (based on square root transformed data) was comparable, with about 80 percent of

the binary data correctly predicted. However, in addition the RF model also provides information about impact severity. The machine learning algorithm seems to be most capable of fitting "difficult" data points. For example, water supply related impacts may persist because of low groundwater levels, despite shorter-term wet conditions. These cases manifest themselves as high observed $N_I$ for very negative values of SPEI-24, but positive or only slightly negative values of SPI-6 (see Figure 2).

The HM showed the lowest predictive performance regarding both the binary and the count parts, with both frequent false alarms and underprediction of high impact counts. One could argue that text-based drought impact information is vaguer or fuzzier than, e.g., data representing ecological processes, where HM was found to be suitable (Ver Hoef and Jansen, 2007; e.g. Potts and Elith, 2006). An increased performance of HM for more conservatively counted impact data (Method 3) supports this speculation. One can infer that for text-based drought impact data non-parametric methods may be most

suitable. Future work could test other machine learning or flexible approaches that have been applied to drought modeling (e.g. Morid et al., 2007). On the other hand, a slight improvement of HM performance by re-assessing the predictor selection may not be ruled out; we do not claim to have identified the optimal model by automatic predictor selection. Nevertheless, small tweaks regarding the in- or exclusion of certain predictors only yielded marginal differences. It can be noted that the study region is very diverse geologically. The SPI duration showing the strongest relationship with monthly mean

streamflow can vary greatly between catchments even over short distances, due to the geological heterogeneity of south-east England (Barker et al., 2016). For most catchments, Barker et al. (2016) found the correlation with streamflow to be strongest for SPI durations less than a year, but for very permeable catchments with a large groundwater contribution to flows, correlations remained strong up to the longest duration studied: two years. Hence, it seems reasonable to include SPI predictors representing both the fast and the slow response to rainfall (the latter including groundwater as well as streamflow

in permeable catchments).

The between-model differences discussed above also apply when subsetting the total impacts on water supply and freshwater ecosystem impacts, respectively. We expected that using subsets of the total impacts would lead to more homogeneous data and thus a closer relation between drought intensity and impact occurrence. However, the analysis did not generally support this. Possible explanations include that the rainfall response of streamflow in very permeable catchments (affecting

freshwater ecology) can be as slow as that of groundwater (affecting water supply). Another reason may be that the subsets may result in less representative data than the lumped data. Data-driven models need sufficient data for training. Because of this we limited the development of drought impact functions to impact categories with many data points, and the larger-scale region SEE. The suitability of our methods for constructing local-scale drought impact functions needs further investigation.





For smaller regions there is less data available in the EDII. A previous study found decreased RF performance for regions with lower data availability (Bachmair et al., 2016a).

A potential application for drought impact functions could be an inclusion into drought early warning systems as an additional layer of information supporting hydro-meteorological indicators. If near real-time monitoring of drought impacts is available, as is the case for the US DIR, impact predictions could be supported by impact information of the preceding time steps. Our analysis revealed an increase in predictive power when including such knowledge. Furthermore, impact functions as surrogates for damage functions could be used with hazard scenarios to derive an estimate of risk (e.g. Stoelzle et al., 2014). However, drought impact functions represent a (rather loose) measure of severity; monetary risk estimates could only be derived by coupling them with approaches to quantify the willingness to pay for the restoration of certain (ecosystem) services (Banerjee et al., 2013; Logar and van den Bergh, 2013; Mens et al., 2015). On the other hand, hydro-economic models or engineering approaches could be tested against such empirically derived impact functions. A caveat is that our impact functions do not incorporate dynamics of vulnerability; i.e. the link between hydro-meteorological indicators and impacts may change over time due to adaption and preparedness measures (Blauhut et al., 2015a), for example the increasing resilience of water supply systems to drought. For monetary losses such changes may be accounted for (e.g. by price adjustment (Kron et al., 2012). In our case the variable Y (year) may cater for trends in vulnerability or impact reporting to some extent as suggested by Stagge et al. (2015b). Interestingly, the year is included as a predictor for all the models of freshwater ecosystem impacts, whereas the LG and HM use only SPI of different durations for estimating water supply impacts.

In assessing the most suitable impact function for any application, further evaluation criteria may be useful in addition to the predictive power, such as the capability of extrapolation beyond the training data, interpretability and simplicity of communication, and ease of application. Especially the ability of RF to predict impact occurrence for yet unexperienced drought scenarios needs to be explored. Although the RF method means that complicated relationships between the (many) predictors and the predictand can be incorporated, the fewer predictors used in the LG and HM approaches make interpretation of the link between indicators and impacts more transparent. In the choice of modelling methodology, a balance therefore needs to be struck between these several different criteria.

## 6 Conclusion

This study tested the potential for developing empirical "drought impact functions" based on hydro-meteorological drought indicators and text-based reports on drought impacts as a surrogate variable for drought damage. With a view to transferring the concept of damage functions (widely used in other hazards) to drought, we tested different methods for quantifying text-based information and three data-driven models for linking hazard intensity with the derived drought impact variables for one example region in South-East England. We conclude that although having some limitations, text-based reports on drought impacts can provide useful information for drought risk management. While the conversion of text-based reports




into number of drought impact occurrences is undeniably more uncertain than binary data of presence/absence of impact occurrence, it provides an additional measure of impact severity that was found reasonably predictable. Unlike more commonly used damage functions linking one hazard variable to one particular type of damage, modeling the impacts of the multi-faceted hazard of drought requires several drought indicators (in our case different accumulation periods of SPI and

SPEI). Out of the three models tested, the random forest model generally performed best. While logistic regression and the random forest model showed a similar discriminatory power for binary impact data, the random forest additionally predicts count data and thus information about impact severity. When using subsets of the total impacts (impacts on water supply and impacts on freshwater ecosystems, respectively) similar between-model differences are revealed. While the flexible machine learning algorithm seems most suitable for modeling the complex relation between drought indicators and text-based data,

we do not claim to have generally identified the best model. Instead, our study showcases different methodological approaches to developing drought impact functions based on text-based data, depending on data availability and purpose of analysis.

**Acknowledgements**

This study is an outcome of the Belmont Forum project DrIVER (Drought Impacts: Vulnerability thresholds in monitoring
and Early warning Research). Funding to the project DrIVER by the German Research Foundation DFG under the international Belmont Forum/G8HORC's Freshwater Security programme (project no. STA-632/2-1) is gratefully acknowledged. Financial support for C. Svensson and J. Hannaford within the DrIVER project was provided by the UK Natural Environment Research Council (Grant NE/L010038/1). Financial support for I. Prosdocimi while at CEH was provided by NERC/CEH National Capability funding.

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

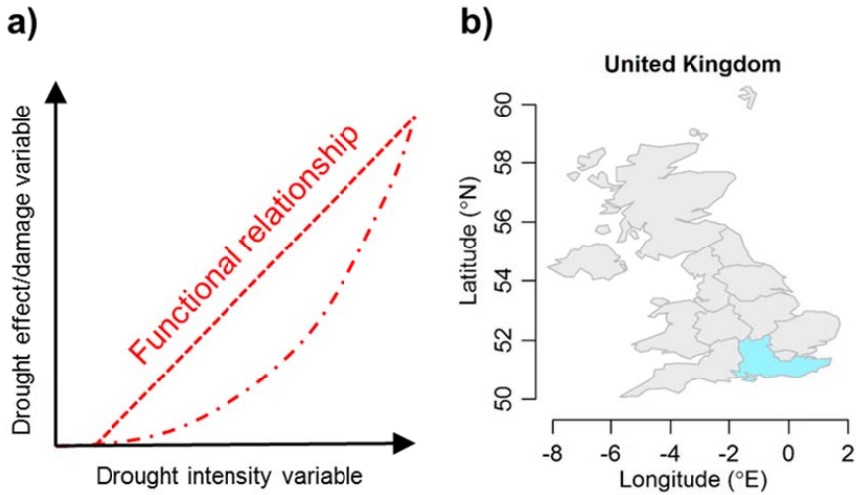

**Figure 1: a) Schematic examples of drought damage functions (red lines), and b) location of the South-East England study area (blue shading) among the NUTS1 regions of the UK.**




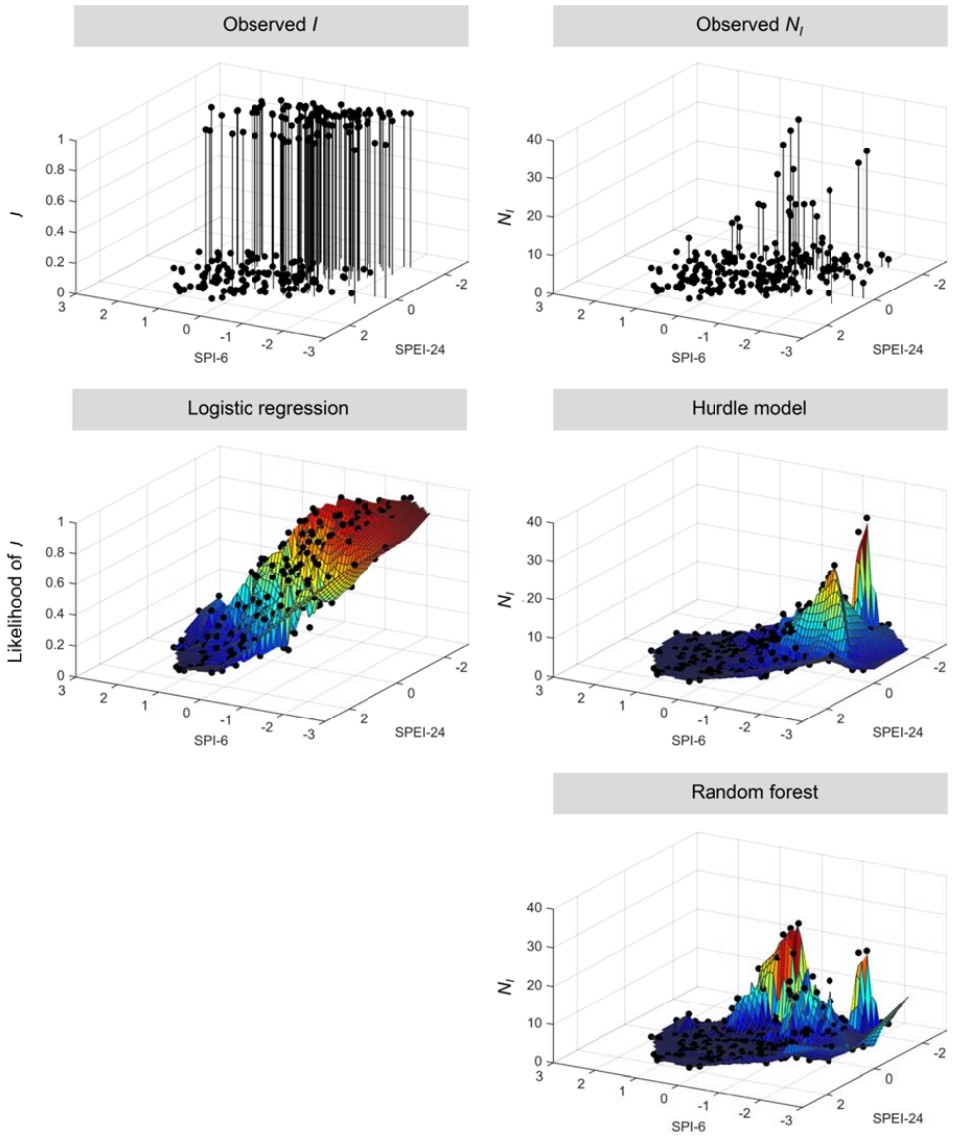

**Figure 2: Top row: dependence of the observed response variable (black dots) on SPI-6 and SPEI-24 (total impacts; NI: impact quantification method 3). Bottom rows: fitted models (only SPI-6 and SPEI-24 are displayed although the models include further predictors).**





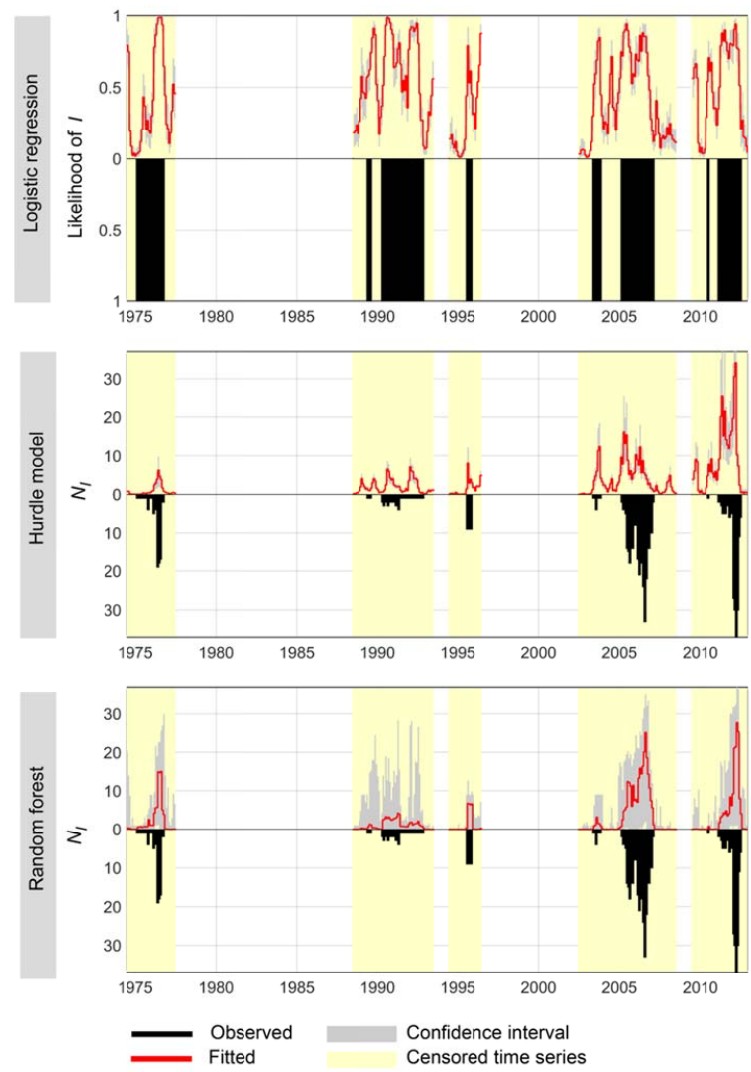

**Figure 3**: **Observed versus fitted time series of *I* or *$N_I$* (total impacts; *$N_I$*: impact quantification method 3).**



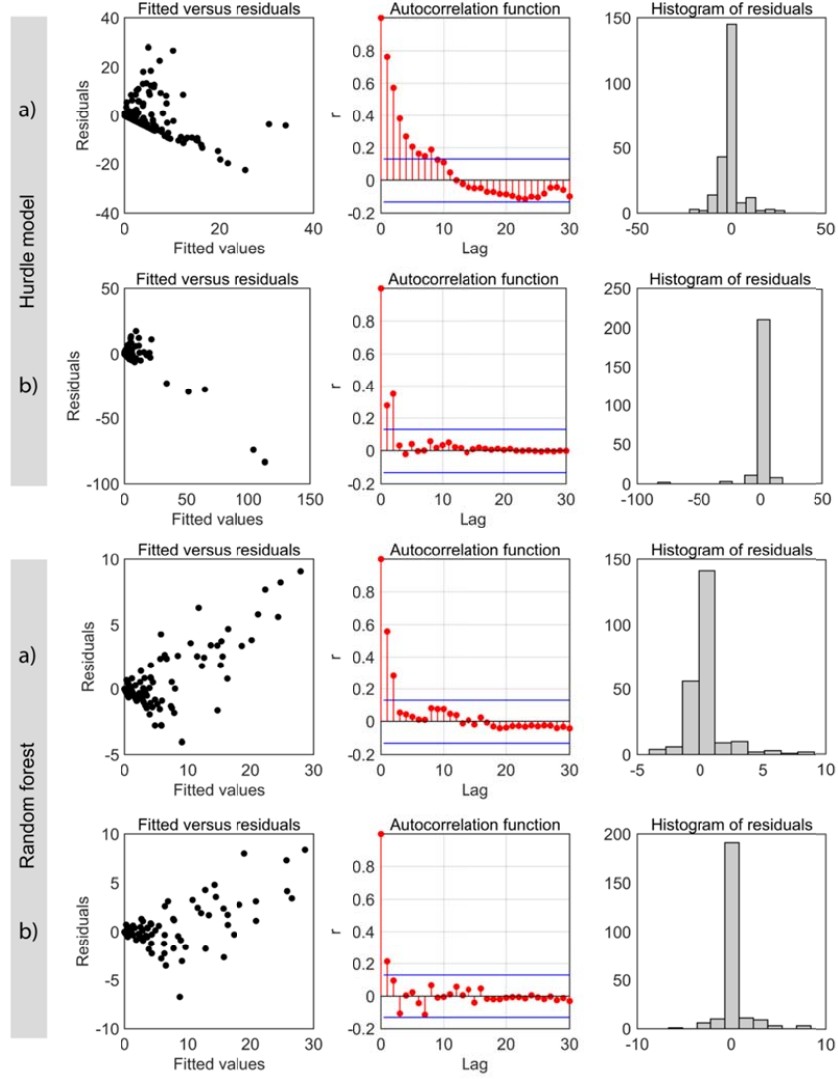

**Figure 4: Raw residuals for both count data models (total impacts; $N_I$ impact quantification method 3). a) Models based on selected predictors (see 3.2); b) $N_I$ of preceding month as additional predictor.**




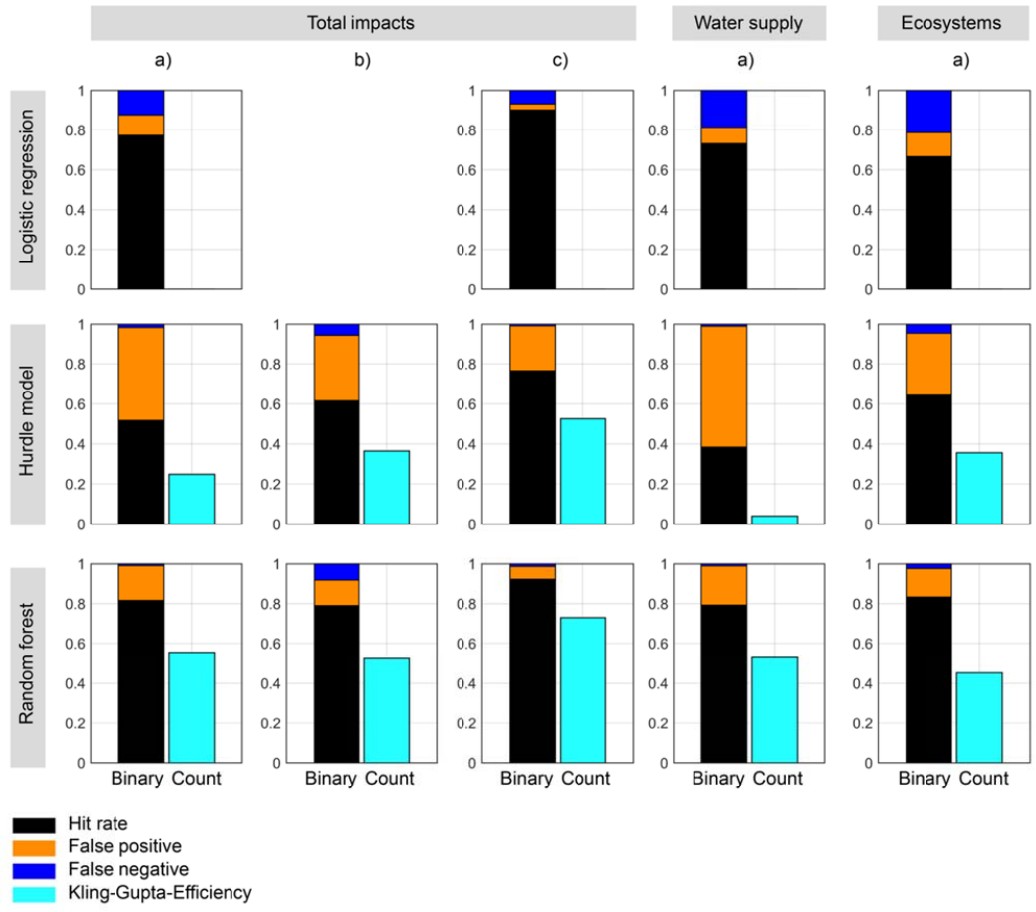

**Figure 5: Model performance metrics based on leave-one-out cross-validation for total impacts and impacts on water supply and freshwater ecosystems. a)** $N_I$ **after impact quantification method 2; b)** $N_I$ **after impact quantification method 3; c) as a) but including** $N_I$ **of preceding month as additional predictor.**





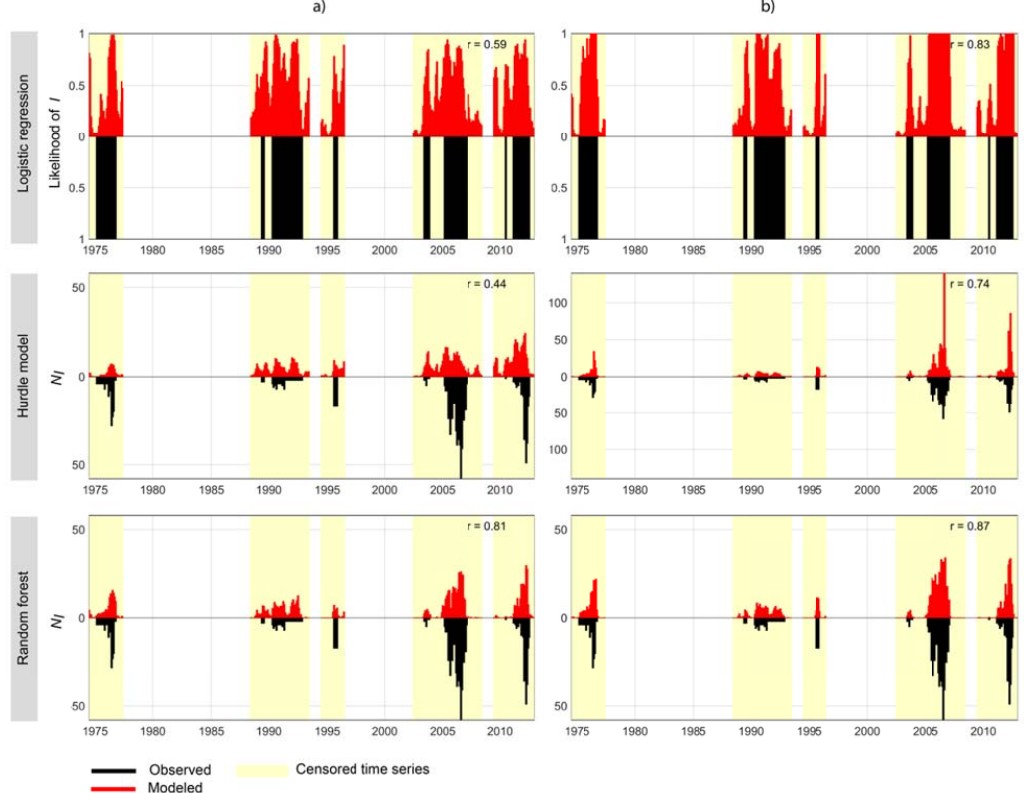

**Figure 6: Examples of observed versus modeled time series based on leave-one-out cross validation.**

a) $N_I$ after impact quantification method 2; b) as a) but including $N_I$ of preceding month as additional predictor.

