# Peer review of "Developing drought impact functions for drought risk management"

_Natural Hazards and Earth System Sciences, 2017_

## Referee Comment (RC1) · Anonymous Referee #1 · 31 Jul 2017

The authors propose an interesting approach to derive "drought impact functions" from text-based reports and assess the possibilities and limitations of transferring these into drought management. The presentation of the work is excellent, with clear objectives, easy to follow methodology and straightforward results. I consider the article should be accepted in its current form as it represents a novel contribution in the field of drought impacts assessment, very valuable for preparedness. My only concern is the limited applicability of the methodology in reality. As the authors acknowledge, a caveat of the impact functions is that they do not incorporate the dynamics of vulnerability. This is, they cannot capture the implementation of adaptation and preparedness measures usually adopted after (or during) a drought event. Therefore, any derived function would only be useful within a drought early warning system if no adaptation measures were
ever adopted, irrespective of the severity of the impacts. Moreover, assuming that the impact functions must be sector specific, the availability of data to derive impact functions must be strongly biased towards one of two sectors. How do the authors expect to cope with scarce data availability? How would the authors expect the impact function concept translates into water resources planning and management?

---

## Referee Comment (RC2) · Anonymous Referee #2 · 2 Aug 2017

This study tests the potential for developing empirical drought impact functions based on drought indicators as predictors, and text based reports as variables for drought damage using South East England as a case study. The study shows that text-based reports can provide valuable information for drought risk management and that various methodological approaches can be applied to develop drought impact functions.

By showcasing and evaluating methodologies for setting up drought impact functions the study provides a valuable addition to the "drought impact debate". Apart from that the study reads well. I would therefore support the manuscript for publication but with substantial revisions taking into account the following general and technical comments/suggestions:

General comments:

[Figure]

1. The study currently only looks at counts of impacts and not the actual height of impact. I wonder whether the authors checked the relation between the count and actual height of impacts, whether this relation is positive of negative. Could the authors elaborate further on this and on the question what would happen with the impact functions if height of impact is taken into account?

2. The authors fit "damage functions" based on a 'leave-on-out' principle. To me it isn't a surprise that with such an approach high correlations/good results are being found. I'd suggest the authors to do some extra sensitivity testing on this issue: e.g. leaving out more variables in the fitting. Could the authors elaborate more on how stable this relation is then? Up to what level (# of points left out) are still reasonable results achieved?

3. From reading the methods it does not become clear to me how exactly you coupled a gridded product (SPI/SPEI) to counted impacts over the basin in SSE. And this you use an equal time-period to establish the fits for the different accumulation times? Please elaborate further on this.

4. Some parts in the methods section are repetitive, for example the paragraphs on P.6. Please take a look and move repetitive sections of text.

Minor comments:

1. P1.L24: "lowest prediction uncertainty": How about the uncertainty when modelling full counts? But using only binary outputs?

2. P1.L25: "Reasonable limits": what limits?

3. P2.L29: "water use... fish kills": As an additional reference, use Logar et al. Costs of drought

4. P3.L2: "vegetation stress": Could you give some more examples here?

5. P3.L5: "rainfall": I'd put this broader: particularly other hydrological variables cause

droughts, e.g. runoff. Especially in an area with relatively high irrigated agriculture

6. P3.L28: "Predictive power": elaborate further on what you mean with this.

7. P4.L1: "SSE": the abbreviation for South-East England is not consistent, see also L5.

8. P5.L4: "drought impact occurrence during each month": Is this a valid assumption? And did you do some (sensitivity) testing to back-up this assumption?

9. P5.L10: "All impact reports are counted": Does it matter in terms of actual height of impact to define and sum these sub-types? 7 sub-types together may have a lower impact than 1 sub-type impact elsewhere.

10. P7.L6: "0.7": Was this value based on literature or randomly chosen?

11. P7.L12-L17: I wonder whether you can do a proper regression for different sectors (and sometimes adding them up) while leaving out years without a drought. In doing so you miss part of the impact mechanism, i.e. droughts without an impact. And shouldn't it be more transparent to use the same n-months of drought for all impact categories?

12. P7.L29: "SPI-6...": Could you elaborate a bit further on whether taking these variables make sense from a physical point of view? And how about double-counting of drought mechanisms? Please also clarify where M and Y stands for, so far not explained in the text yet.

13. P8.L10: "Positive impacts": Do you mean here, an impact measured?

14. P8.L10: "Four instances": is it possible to highlight these with a different color. From looking at the figure I count more than 4.

15. P8.L11: "yet negative indicator values": I can image that this can be explained by demand/supply mechanisms. Did you check for which impact indicators this holds?

16. P8.L15: "The RF model better captures": How can we see this from the figure?

17. P8.L16: "SPEI-24 but less negative": Did you also compare the performance of indicators when using only SPI-6 or SPI-24?

18. P8.L28: "two data-points": Could you point this out in the figures. Not completely clear to which points you refer here.

19. P9.L1-L4: Specify where we can find these results, e.g. in figures or tables

20. P9.L13: 'including impact information from the preceding month": How did you do this exactly? And are there many one-month impacts or are most months characterized by multi-month impacts?

21. P10.L20: "Streamflow": I don't exactly understand where the term streamflow comes from here. Please elaborate.

22. P11.L21: "to predict impact occurrence for yet unexperienced drought scenarios". I wonder whether this methods can be used for extrapolation or prediction of impacts for unexperienced drought scenarios, especially in case these drought scenarios become more extreme. By using SPI/SPEI values the drought range was more or less fixed from -3 up to +3 (SPI/SPEI values), representing the variability from a long-term mean state. I wonder how you would deal with e.g. the increased severity of droughts. What if the value behind a -3 would become a -2 or -4 when taking a different/longer time-series into account, e.g. due to the effects of climate change? Would it be an option to add a trend to impacts in order to account for this issue?

23. P11.L29: "damage functions": Damage functions are often used for a specific grid, i.e. a pixel flooded, resulting in a certain degree of (monetary) damage. I'm not totally sure whether it is appropriate to call this damage functions when being applied on the scale of a basin or when including only counts as measure of severity. How would you interpret "maximum damage" here, or "vulnerability"?

24. P21.Fig5: Could you elaborate further on how to interpret these figures given the leave-on-out cross validation that was performed? From looking at this figure I understand that the hurdle approach underestimates impacts. Could you discuss/elaborate more whether it would be more important for a policy maker to under- or overestimate impacts, from example from a political or safety point of view?

———————————————————

---

## Author Response (AR1)

**Review 1**

Thank you to reviewer 1 for the supportive feedback and pointing out issues that require more discussion in the manuscript. The reviewer's concern mainly centers on the applicability of the methodology in reality: *"My only concern is the limited applicability of the methodology in reality. As the authors acknowledge, a caveat of the impact functions is that they do not incorporate the dynamics of vulnerability."*
Further posed questions are: *"How do the authors expect to cope with scarce data availability? How would the authors expect the impact function concept translates into water resources planning and management?"*

We added content to the Discussion section of the paper on the topics outlined below to put the presented methods into a broader context.

1) Applicability in reality, particularly to water resources planning and management

We acknowledge that there are still some hurdles towards an operational applicability, and that these could be better discussed. We will expand the current discussion paragraph on the applicability of our concept for monitoring and early warning purposes (see page 11 lines 3-18). In general, this being a science paper rather than a guideline for operational use, we hope that the proposal of a new concept will find further test beds and provide a seed for further developments.

As we point out in the above mentioned discussion paragraph, drought impact functions may provide different usage options. Generally, they may be applied for planning and risk management or in a real-time, early warning context. For the case of water resources planning and management, in particular, scenarios provide a common tool to test and improve existing drought plans. Currently such tests are mostly done at the level of individual water suppliers with very specific failure functions. An application of the impact functions may allow a more regional to country scale assessment of the risk of certain sectors to drought and hence enable the improvement of emergency plans across sectors.

In a real time context, an impact function will allow to interpret the given monitoring of drought indices as a threshold or trigger of action. Impact functions thereby translate drought intensity expressed by a hydro-meteorological indicator into the possibility of experiencing socio-economic or ecological effects. Since the impact functions are based on historical experience, they can be interpreted as a warning that a certain currently experienced drought condition resulted in negative effects on specific sectors in the past. This information could potentially be used to trigger management actions. Currently, operational systems like the US Drought Monitor or the European Drought Observatory allow the user to select different drought indices. We propose that in the future users should ideally also be able to select an index that provides information on whether socio-economic or ecological effects can be expected for this drought intensity, which could be addressed by impact functions. This would go some way to countering a recognized deficiency in M&EW systems, i.e. a capacity to quantify and eventually predict impacts on society and ecosystems (Bachmair et al. 2016 in reference list).

2) Dynamics of vulnerability

As correctly pointed out, the dynamics of vulnerability play an important role when designing impact functions. We included the year of impact occurrence as additional variable that may cater for trends in vulnerability or impact reporting to some extent as suggested by Stagge et al. (2015b). To our knowledge, the issue of dynamics of vulnerability also applies to other natural hazards impact or damage functions and is not trivial. If a drought impact function is designed for a specific application

and region, expert elicitation could be used to gain an understanding of dynamics of adaption measures over time. One would need to test whether quantifying such information and adding it as further predictor variable would improve the reliability of impact functions. Further approaches for considering the dynamics of vulnerability are presented by Blauhut et al. (2015a). We have added text to this effect to the Discussion (page 12).

3) Data availability

To overcome the data scarcity, a number of suggestions have been made and appear possible given the will and resources. Options include semi-automated newspaper clippings and other big data approaches as well as trained observer networks as explored with the USA drought monitoring (see e.g. Smith et al. (2014). They also include a more directed and targeted monitoring of drought impacts in general as part of environmental and water resources monitoring. We have added text to this effect to the Discussion (page 11).

**Review 2**

Thank you for the detailed and constructive feedback to our study. We appreciate all minor editorial and clarification comments.

**Reply to major/general comments:**

*"1. The study currently only looks at counts of impacts and not the actual height of impact. I wonder whether the authors checked the relation between the count and actual height of impacts, whether this relation is positive of negative. Could the authors elaborate further on this and on the question what would happen with the impact functions if height of impact is taken into account?"*

We presume that by "height of impact" the reviewer means some numerical measure of impact severity. This type of information is not available in the text-based drought impact reports per se. We use different methods of quantifying the coded drought impacts: one is the use of binary impact information (presence versus absence of an impact per month) and another is the count how many impacts were reported for a given month. The latter method considered different ways of counting (see methods section 2.3). By counting the number of impacts reported we obtain a measure of impact severity, although this reflects the comprehensiveness of the drought impacts rather than relating a particular value of the SPI/SPEI to, say, a particular value of agricultural yield loss, or similar. Yield data would provide an objective measure of impact severity, but it would only reflect a very limited range of drought impact types.

Other options to assign a severity measure to impact data were not found reasonable. The EDII database provides impact report data that are based on text reports and coded into a very detailed system of impact categories and subtypes. The subtypes in some, but not in all categories may represent different levels of severity of an impact (for water supply they range from awareness, to bans, to actual supply restrictions, for example), as discussed in Stahl et al. (2016). However, there is no additional severity coding in the EDII database, i.e. no information about impact severity in a standardized/objective way is currently available from the database (see Stahl et al. (2016)).

During early stages of this project we in fact conducted a small test, asking a group of people to rate the severity of selected drought impact reports according to severity classes (low-medium-high). This small test revealed the complexity of how impact severity is perceived depending on the impact category, knowledge about drought impacts, affected area, and level of detail in the report. Given the subjectivity of text-based data that are available, we used different impact quantification methods, which have not been addressed so far. Exploring different methods for counting the number of impact reports as we did in our study is in our opinion the best way currently to somehow address impact severity. We clarified this in the paper (see new sentences on page 10).

*"2. The authors fit "damage functions" based on a 'leave-on-out' principle. To me it isn't a surprise that with such an approach high correlations/good results are being found. I'd suggest the authors to do some extra sensitivity testing on this issue: e.g. leaving out more variables in the fitting. Could the authors elaborate more on how stable this relation is then? Up to what level (# of points left out) are still reasonable results achieved?"*

The aim of the study was to test and compare three data-driven models for linking drought intensity with drought impacts, applying each model in the best possible way with regard to 1) number of data points for fitting, and 2) choice of predictor variables. The reviewer seems to be concerned about both issues, i.e. number of data points for fitting (addressed in this comment) and selection of predictor variables (see minor comment 12 about *"Could you elaborate a bit further on whether taking these variables make sense from a physical point of view? And how about double-counting of drought mechanisms?"*).

Number of data points for fitting:
Leave-one-out cross-validation is a very common approach. Given the issue of impact data scarcity, we think that we should use the largest dataset possible for model fitting. Clearly, less data points for fitting will deteriorate results. If our focus was on designing impact functions for operational use, we agree that further testing of the effect of sample size would be needed. However, for the purpose of comparing different approaches for drought impact functions (i.e. relative to each other) in our opinion a leave-one-out cross-validation is appropriate.

Choice of predictor variables:
The selected predictor variables for each model make sense from a physical point of view. In most models a combination of shorter-term and longer-term time scales of SPI or SPEI was selected, and the month or year of impact occurrence (the variables M and Y are introduced on P4 L 20-22). While for the RF model all predictors are used, the above-named ones were also identified as the most important ones. Finding and interpreting 'best-predictors', i.e. physical indicator relating most strongly to impact occurrence, has been the focus of preceding studies, e.g. Stagge et al. (2015b) or Blauhut et al. (2016). The aim here was not to repeat again the previous results like 'longer duration water deficits relate more to water resources impacts because of longer response times of such systems', which have been previously shown. The aim was to compare the different methodological approaches and to initiate the idea to work towards potential impact function derivation. Nevertheless, we agree that a short paragraph on the differences of the selected predictors will likely be useful to the reader and we suggest to add this into the discussion section. We are not entirely sure what is meant by 'double counting'. Possibly the interrelation of predictors in a multiple predictor model? In our view this is not an issue given the way we selected the predictors excluding highly correlated ones (P 7 L 5-7).

*"3. From reading the methods it does not become clear to me how exactly you coupled a gridded product (SPI/SPEI) to counted impacts over the basin in SSE. And this you use an equal time-period to establish the fits for the different accumulation times? Please elaborate further on this."*

Thank you for pointing out that this is currently not clearly described. For each month we calculated the regional average of all E-OBS grid cells falling within the polygon covering South East England. The regional average was chosen since Bachmair et al. (2015) found little difference between the performances of different regional indicator metrics (e.g. mean vs. minimum vs. maximum etc.). The SPI/SPEI accumulation durations reflect the water deficit accumulated in the SEE area over that duration, and we relate this to the number of impacts occurring in the single month following the SPI/SPEI accumulation duration.  We added this information to the Methods section 2.2.

*4. Some parts in the methods section are repetitive, for example the paragraphs on*
*P.6. Please take a look and move repetitive sections of text.*
→ We carefully read through page 6 but do not see any repetitive sections.

**Reply to minor comments:**

1. P1.L24: "lowest prediction uncertainty": How about the uncertainty when modelling full counts? But using only binary outputs?
→ We conducted exactly this suggested analysis (modelling full counts but evaluating the binary output separately from the count output). Please see the Methods section (P7 L19-23).

2. P1.L25: "Reasonable limits": what limits?
→ We changed this to "were found to be reasonably predictable". We meant this in a more qualitative way rather than intending to provide limit values.

3. P2.L29: "water use: : : fish kills": As an additional reference, use Logar et al. Costs of drought
→ We added this reference.

4. P3.L2: "vegetation stress": Could you give some more examples here?
→ We added some more examples that were investigated in the cited papers.

5. P3.L5: "rainfall": I'd put this broader: particularly other hydrological variables cause droughts, e.g. runoff. Especially in an area with relatively high irrigated agriculture
→ The most commonly used drought indicators are rainfall based indicators, as found by Bachmair et al. (2016) based on survey targeted on drought monitoring and early warning systems. Streamflow and further indicators like soil moisture or groundwater levels are (unfortunately) mostly not as easily available as rainfall. Therefore we would like to keep the sentence with a focus on rainfall.

6. P3.L28: "Predictive power": elaborate further on what you mean with this.
→ We think this a commonly used term. Nevertheless, we changed it to "predictive performance", which may be clearer.

7. P4.L1: "SSE": the abbreviation for South-East England is not consistent, see also L5.
→ Thank you for noticing this. We consistently changed it to SEE.

8. P5.L4: "drought impact occurrence during each month": Is this a valid assumption? And did you do some (sensitivity) testing to back-up this assumption?
→ One of the great challenges of working with text-based data is how to adequately transform this into quantitative data. To our knowledge, our study is the first one to test the effect of some of these assumptions (see different ways of impact counting described on page 5). We agree that your suggestion to test the effect of the assumptions regarding the temporal occurrence would be very interesting to address in future research. We added the following sentence on P10, L2-3: "Further sensitivity tests on the assumptions during impact quantification are desirable future research, e.g. testing the effect of assigning a time of occurrence when impact reports only provide an approximate time indication."

9. P5.L10: "All impact reports are counted": Does it matter in terms of actual height of impact to define and sum these sub-types? 7 sub-types together may have a lower impact than 1 sub-type impact elsewhere.
→ Please see out reply to major comment 1 on "impact height", minor comment 9, and the details in the Methods section 2.3. We investigated the effect of how to count impact subtypes. However, since a measure of actual, objective impact severity is not available to us, 7 sub-types will not be lower than 1 impact subtype elsewhere.

10. P7.L6: "0.7": Was this value based on literature or randomly chosen?

→ This value was chosen based on a mixture of prior data investigation and literature. We did not systematically assess the sensitivity of this value but during the data analysis this value appeared to be a good cut-off point. As also pointed out in the reply to the major comment on the choice of predictor variables, the aim here was not to repeat again results of previous studies on 'best predictors' for certain drought impact types. The aim was to compare the different methodological approaches and to initiate the idea to work towards potential impact function derivation.

11. P7.L12-L17: I wonder whether you can do a proper regression for different sectors (and sometimes adding them up) while leaving out years without a drought. In doing so you miss part of the impact mechanism, i.e. droughts without an impact. And shouldn't it be more transparent to use the same n-months of drought for all impact categories?

→ As stated in the manuscript the rationale for this is that there may be a lack of impact reporting for certain drought events; hence we only focus on parts of the time series with reported drought impacts. All months of all years with drought impact occurrence were selected plus an additional six months buffer before and after the drought year to include sufficient variability for model training. We have discussed this also in a previous paper in more detail. Please refer to Bachmair et al (2015) for details.

12. P7.L29: "SPI-6: : :": Could you elaborate a bit further on whether taking these variables make sense from a physical point of view? And how about double-counting of drought mechanisms? Please also clarify where M and Y stands for, so far not explained in the text yet.

→ Please see reply to major comment 3, where we address this issue. The predictors M and Y are explained on P4, L24-25. "As additional predictors, used to account for temporal trend and seasonality, we chose the year (Y) and the month (M, expressed as a sinusoid) of impact occurrence (Bachmair et al., 2016a)."

13. P8.L10: "Positive impacts": Do you mean here, an impact measured?
→ Yes. However, we removed the word 'positive' and changed it to just 'impact counts' since this term seemed to be misleading.

14. P8.L10: "Four instances": is it possible to highlight these with a different color. From looking at the figure I count more than 4.
→ We prefer to keep the figures as they are and not add more complexity that needs extra labeling and description. We also think that referring to the 'front left quadrant' as we do in the manuscript should guide the reader. However, we replaced "4" by "a few" just to avoid confusion when interpreting the figure. It is indeed a bit hard to discern the exact location of the points within the 3-D-plots.

15. P8.L11: "yet negative indicator values": I can image that this can be explained by demand/supply mechanisms. Did you check for which impact indicators this holds?

→ In the Discussion section on page 10 we speculate that "For example, water supply related impacts may persist because of low groundwater levels, despite shorter-term wet conditions." We believe that this what reviewer 2 had in mind when suggesting demand/supply mechanisms. The above named impacts indeed predominantly relate to water supply impacts.

16. P8.L15: "The RF model better captures": How can we see this from the figure?
→ One can see this by visually comparing the pattern of observed impact counts (top right panel) to the modeled impact counts (lower panels). Especially since reviewer 1 did not criticize this we prefer to not add extra description on this to the text.

17. P8.L16: "SPEI-24 but less negative": Did you also compare the performance of indicators when using only SPI-6 or SPI-24?
→ No. The random forest model is based on all predictors, as described in the Methods section. Best performing predictors are identified within the algorithm. The SPI-24 values for the highlighted data points are, however, very similar to the SPEI-24 values and therefore a similar pattern holds true.

18. P8.L28: "two data-points": Could you point this out in the figures. Not completely clear to which points you refer here.
→ We added the following reference to the sentence: "(see HM b) in Figure 4)"

19. P9.L1-L4: Specify where we can find these results, e.g. in figures or tables
→ We now refer to Figure 5 in the text and hope this helps to guide interpretations.

20. P9.L13: 'including impact information from the preceding month": How did you do this exactly? And are there many one-month impacts or are most months characterized by multi-month impacts?
→ The impact data was prepared as described in section 2.3. The impact information from the previous month was included as an extra predictor variable, which we are now clarifying in the text. There are both one-month and multi-months-impacts in the time series depending on the year and impact category.

21. P10.L20: "Streamflow": I don't exactly understand where the term streamflow comes from here. Please elaborate.
→ We have now clarified that the geology affects the response time of river flows (i.e. streamflows) to rainfall.

22. P11.L21: "to predict impact occurrence for yet unexperienced drought scenarios". I wonder whether this methods can be used for extrapolation or prediction of impacts for unexperienced drought scenarios, especially in case these drought scenarios become more extreme. By using SPI/SPEI values the drought range was more or less fixed from -3 up to +3 (SPI/SPEI values), representing the variability from a long-term mean state. I wonder how you would deal with e.g. the increased severity of droughts. What if the value behind a -3 would become a -2 or -4 when taking a different/longer timeseries into account, e.g. due to the effects of climate change? Would it be an option to add a trend to impacts in order to account for this issue?
→ The reviewer is certainly right that data-driven models have their limitations regarding unprecedented drought condition, and that any tentative extrapolation would need to be done with full disclosure of that the model has not been calibrated for these extreme values. We already have year as an optional predictor, which accounts for any trend in the time span of the observations.

23. P11.L29: "damage functions": Damage functions are often used for a specific grid, i.e. a pixel flooded, resulting in a certain degree of (monetary) damage. I'm not totally sure whether it is appropriate to call this damage functions when being applied on the scale of a basin or when including only counts as measure of severity. How would you interpret "maximum damage" here, or "vulnerability"?
→ We agree that the term 'damage functions' may not be applicable to our outlined concept. This is exactly the reason why we use the term 'drought impact' function instead.

24. P21.Fig5: Could you elaborate further on how to interpret these figures given the leave-on-out cross validation that was performed? From looking at this figure I under-stand that the hurdle approach underestimates impacts. Could you discuss/elaborate more whether it would be more

important for a policy maker to under- or overestimate impacts, from example from a political or safety point of view?

→ Yes, high impact counts are underestimated. We describe this on page 5. "The HM shows an even stronger underprediction of high NI and frequent impact occurrence predictions despite absent impacts, resulting in KGE values less than 0.4." Regarding the second part of the questions it depends. Underestimation of severe consequences is, of course, more dangerous. Nevertheless, "false positive alarms" without impacts following this are also problematic regarding public awareness and willingness to prepare.